# NOX4 Mediates *Pseudomonas aeruginosa*-Induced Nuclear Reactive Oxygen Species Generation and Chromatin Remodeling in Lung Epithelium

**DOI:** 10.3390/antiox10030477

**Published:** 2021-03-17

**Authors:** Panfeng Fu, Ramaswamy Ramchandran, Tara Sudhadevi, Prasanth P. K. Kumar, Yashaswin Krishnan, Yuru Liu, Yutong Zhao, Narasimham L. Parinandi, Anantha Harijith, Junichi Sadoshima, Viswanathan Natarajan

**Affiliations:** 1Departments of Pharmacology & Regenerative Medicine, University of Illinois at Chicago, Chicago, IL 60612, USA; panfengfu@hotmail.com (P.F.); ramchan@uic.edu (R.R.); punathil@uic.edu (P.P.K.K.); ykrish2@uic.edu (Y.K.); yuruliu@uic.edu (Y.L.); 2Department of Pediatrics, Case Western Reserve University, Cleveland, OH 44106, USA; txs671@case.edu (T.S.); axh775@case.edu (A.H.); 3Department of Physiology and Cell Biology, Ohio State University, Columbus, OH 43210, USA; yutong.zhao@osumc.edu; 4Department of Internal Medicine, Ohio State University, Columbus, OH 43210, USA; narasimham.parinandi@osumc.edu; 5Department of Cell Biology & Molecular Medicine, Rutgers New Jersey Medical School, Newark, NJ 07103, USA; sadoshju@njms.rutgers.edu; 6Department of Medicine, University of Illinois, Room 3137 COMRB Building 909, South Wolcott Avenue, Chicago, IL 60612, USA

**Keywords:** *Pseudomonas aeruginosa* infection, NOX4, nuclear ROS, histone acetylation, HDAC1/2 oxidation, lung epithelium

## Abstract

*Pseudomonas aeruginosa* (*PA*) infection increases reactive oxygen species (ROS), and earlier, we have shown a role for NADPH oxidase-derived ROS in *PA*-mediated lung inflammation and injury. Here, we show a role for the lung epithelial cell (LEpC) NOX4 in *PA*-mediated chromatin remodeling and lung inflammation. Intratracheal administration of *PA* to Nox4^flox/flox^ mice for 24 h caused lung inflammatory injury; however, epithelial cell-deleted Nox4 mice exhibited reduced lung inflammatory injury, oxidative stress, secretion of pro-inflammatory cytokines, and decreased histone acetylation. In LEpCs, NOX4 was localized both in the cytoplasmic and nuclear fractions, and *PA* stimulation increased the nuclear NOX4 expression and ROS production. Downregulation or inhibition of NOX4 and PKC δ attenuated the *PA*-induced nuclear ROS. *PA*-induced histone acetylation was attenuated by *Nox4*-specific siRNA, unlike *Nox2*. *PA* stimulation increased HDAC1/2 oxidation and reduced HDAC1/2 activity. The *PA*-induced oxidation of HDAC2 was attenuated by *N*-acetyl-L-cysteine and siRNA specific for *Pkc δ*, *Sphk2*, and *Nox4*. *PA* stimulated RAC1 activation in the nucleus and enhanced the association between HDAC2 and RAC1, p-PKC δ, and NOX4 in LEpCs. Our results revealed a critical role for the alveolar epithelial NOX4 in mediating *PA*-induced lung inflammatory injury via nuclear ROS generation, HDAC1/2 oxidation, and chromatin remodeling.

## 1. Introduction

*Pseudomonas aeruginosa* is a common opportunistic Gram-negative bacterium that causes serious life-threatening and nosocomial infection, including pneumonia. *P. aeruginosa* infection is associated with significant morbidity and mortality, prevalent with immuno-compromised patients and those with severe burns, diabetes, cancer, organ transplant, cystic fibrosis, chronic obstructive pulmonary disease, and on mechanical ventilation support [1,2,3,4,5,6,7]. *P. aeruginosa* infection of the lung alters the host genome to facilitate its replication and virulence and also initiates cascade of events in the host, including innate immune responses, reactive oxygen species (ROS) generation, cytokine production, inflammation, dysregulated lipid metabolism, and modulation of epigenetic factors [8,9,10,11]. Recent studies strongly suggest the regulation of sphingolipid metabolism and signaling pathways by *P. aeruginosa* infection of the mouse lung. Decreased sphingosine and elevated ceramide levels have been reported in respiratory tract and airway epithelium of cystic fibrosis patients infected with *P. aeruginosa* compared to normal subjects [12], and sphingosine instillation attenuated the *P. aeruginosa* infection of mouse lung [13]. Further, *P. aeruginosa*-mediated inflammatory lung injury was reduced in mouse with genetic deletion of sphingosine kinase 2 (*Sphk2*) but not *Sphk1*, and by inhibition of SPHK2 with the small molecule inhibitor, ABC294640 [14]. Moreover, *P. aeruginosa* infection of the mouse lungs and epithelial cells modulated histone H3 and H4 acetylation via protein kinase C (PKC) δ-dependent nuclear SPHK2 phosphorylation and S1P production [14].

Several inflammatory response genes including monocyte chemotactic protein (MCP-1), C-C motif chemokine ligand 2 (CCL2), tumor necrosis factor (TNF) -α-induced protein A20 (TNFAIP3), nuclear factor kappa-light-chain-enhancer of activated B cell (NF-κB), and E74-Like ETS Transcription factor (Elf) 3, were upregulated in response to *P. aeruginosa* infection of the mouse lung [8,10,11]. Interestingly, genetic deletion of *Sphk2* attenuated *P. aeruginosa*-mediated activation of many of the aforementioned pro-inflammatory genes belonging to the NF-κB pathway of inflammation in the mouse lung [14]. Further, instillation of *P. aeruginosa* in airways stimulated NF-κB in epithelial cells much earlier compared to other cell types, and enhanced nicotine adenine dinucleotide phosphate (NADPH) oxidase (NOX) 2 and 4 expression in murine lung tissues and human lung endothelial cells via the NF-κB pathway [15]. Further, deletion of NOX2 and NOX4 in the murine lung by siRNA significantly reduced oxidative stress and inflammatory response to *P. aeruginosa* [15].

While a role for the endothelium has been clearly established in NOX2/NOX4-dependent oxidative stress and lung inflammation in response to *P. aeruginosa* [15,16], the contribution of the alveolar epithelial NOX4-dependent lung inflammatory injury is unknown. Further, as *P. aeruginosa* infection of the lung epithelium was shown to stimulate nuclear signaling and epigenetic regulation of interleukin-6 (IL-6) production [13,17], in the current study, we sought to (i) investigate the role of epithelial NOX4 in ROS generation and lung inflammatory injury in murine lung by specific deletion of NOX4 in the epithelium, and (ii) determine the interrelationship between *P. aeruginosa*-mediated nuclear SPHK2/S1P signaling and nuclear NOX4 in ROS production and modulation of histone acetylation via histone deacetylase (HDAC) 1/2 oxidation in the lung epithelium. Our current results show that deletion of *Sphk2*, but not *Sphk1*, reduced *P. aeruginosa*-induced NOX4 expression in the lung, and specific deletion of NOX4 in lung epithelial cells significantly reduced the *P. aeruginosa*-induced lung inflammatory injury and acetylation of H3 and H4 histone in mice. In vitro, *P. aeruginosa* stimulated nuclear ROS generation in lung epithelial cells that was dependent on NOX4, but not NOX2, and deletion of NOX4 with siRNA attenuated *P. aeruginosa*-mediated H3 and H4 histone acetylation, and HDAC activity. Deletion of NOX4, PKC δ, and SPHK2 with siRNA also reduced *P. aeruginosa-*induced oxidation of HDAC2 in lung epithelial cells. Our results also show that nuclear SPHK2/S1P signaling regulates NOX4-dependent nuclear ROS via Rac1.

## 2. Materials and Methods

### 2.1. Animals and Animal Procedures

All experimental procedures were approved by the Institutional Animal Care Committee of the University of Illinois at Chicago (UIC). *Nox4^f/f^* mice used in this study were described previously [18], and tamoxifen-inducible Spc-Cre mice (Sftpc-CreER^T2^) were obtained from Jackson Laboratories (Bar Harbor, ME, USA). The mice were bred at the UIC BRL facility and were maintained in a barrier facility with free access to food and water. Both male and female mice, 8 weeks old, were used. Tamoxifen-inducible *Nox4^−/−^* mice (*Nox4^f/f^-Spc-Cre^+/−^* mice) were produced by crossing *Nox4^f/f^* mice with *Spc-Cre^+/+^* mice. Breeding was carried out by crossing *Nox4^f/f^* mice with *Nox4^f/f^-Spc-Cre^+/+^*, with a 50% yield of *Nox4^f/f^-Spc-Cre*-negative and *Nox4^f/f^-Spc-Cre^+/−^* animals. Genetic deletion of *Nox4* in *Nox4^f/f^-Spc-Cre^+/−^* mice was induced by intraperitoneal (i.p.) injection of tamoxifen (200 mg/kg dissolved in autoclaved corn oil) for 5 consecutive days and used for experiments 15 days later. *Nox4^f/f^* mice serving as controls also received the same dosage of tamoxifen via i.p. injection for 5 consecutive days and were experimented 15 days post-tamoxifen injection. Mice were anesthetized with ketamine (100 mg/kg), xylazine (5 mg/kg), and intratracheally administered 20 μL of PBS or live *PA*103 (1 × 10^6^ CFU/mouse) prepared in 20 μL of PBS as described [14,15]. Dorsal sub-cutaneous (s.c.) injection of buprenorphine (0.1 mg/kg) was applied for post-operative analgesia. After 24 h of treatment, bronchoalveolar lavage (BAL) fluids were collected using 1 mL of sterile Hanks Balanced Salt Buffer. Cytokines in the BAL fluids were measured using enzyme-linked immuno-absorbent assay (ELISA), and H_2_O_2_ by Amplex Red Hydrogen Peroxide/Peroxidase kit (Invitrogen, Carlsbad, CA, USA). Left lungs were removed and fixed with 10% formalin, and paraffin-embedded 5 μm thick sections were stained with hematoxylin and eosin for immunohistochemistry. Right lung was snap-frozen in liquid nitrogen and stored at −80 °C for the preparation of total lysates and Western blot analysis.

### 2.2. Isolation and Culture of Mouse Lung Alveolar Type II Cells

Tamoxifen-treated *Nox4^f/f^-Spc-Cre^+/−^* mice or *Nox4^f/f^* control mice were euthanized by 10 min CO_2_ incubation and the lungs were perfused with 10 mL PBS through the right ventricle. The lungs were intratracheally injected with 2 mL dispase via a 1 mL catheter and tightened at the trachea with suture. The lungs were removed and incubated in dispase at room temperature for 45 min. Dissociated lung tissue were treated with DNase (100 U/mL) in 5 mL Dulbecco’s Modified Eagle Medium (DMEM) with 10% fetal bovine serum (FBS), 25 mM [4-(2-hydroxyethyl-1-piperazineethanesulfonic acid] (HEPES), 1 × penicillin/streptomycin, and 1 × gentamicin/amphotericin, followed by sequential filtering through a 70 μm cell strainer and 20 μm nylon membrane. Cell suspension was centrifuged at 150× *g* for 8 min. The type II cell pellet was resuspended in 7 mL DMEM/FBS/HEPES solution and plated onto a CD16/32 and CD45 antibodies-coated plate. After 1 h incubation at 37% with 5% CO_2_ supplement, non-adherent type II cells were collected [19]. Red blood cells (RBCs) were removed by applying RBC lysis buffer, followed by washing with DMEM. Cells were then cultured on 0.2% gelatin-coated culture dishes in DMEM supplemented with 10% FBS, 1 mM HEPES, and penicillin/streptomycin [19].

### 2.3. Downregulation of PKC δ, SPHK2, NOX2, and NOX4 Proteins with Small Interfering RNA

Depletion of endogenous PKC δ, SPHK2, and NOX proteins in lung epithelial cells was carried out using gene-specific small interfering RNA (siRNA), as described previously [15,20]. Briefly, pre-designed *Pkc δ*, *Sphk2*, *Nox2*, and *Nox4* or nonspecific/non-targeting siRNA (Santa Cruz Biotech, Dallas, TX, USA) were used to transfect mouse lung epithelial (MLE) 12 cells. Cells (~70% confluence) were starved in DMEM media containing 2% FBS for 2 h prior to transfection with 50 nM scrambled, *Pkc δ*, *Sphk2*, *Nox2, or Nox4* siRNA complexes, which were prepared in Gene Silencer transfection reagent according to the manufacturer’s recommendation (Genlantis, San Diego, CA, USA). Transfected cells were used 72 h post-transfection.

### 2.4. Imaging and Measurement of Nuclear ROS Using HyPer Biosensor in Epithelial Cells

ROS biosensor HyPer targeted to cytoplasm, mitochondria, and nucleus were obtained from Evrogen (Moscow, Russia). Cells grown to ~80% confluence on 35 mm poly-d-lysine-coated glass-bottom dishes were transfected with HyPer plasmid (100 nM) prepared in Gene Silencer transfection reagent as described above for 3 h followed by culturing the cells in complete DMEM supplemented with 10% FBS for 48 h prior to exposure to heat-inactivated *PA* (1 × 10^8^ CFU/mL) for indicated time periods. Live cell ROS detection was performed with a Carl Zeiss 780 Confocal microscope equipped with a temperature-controlled chamber maintained with humidified 5% CO_2_. Thirty minutes after cells were adapted to the chamber environment, basal ROS images were acquired with 63×/1.40 oil objective at excitation wavelength of 488 nm for detection of ROS biosensor. After basal ROS was acquired, *PA* was added into cell culture medium, followed by image acquisition at different time points. Quantification of ROS was performed by scoring at least 20 cells per field and a minimum of 4–6 fields in each dish, and image analysis was performed using Image J, a Java-based image processing program developed at the National Institutes of Health (NIH ImageJ software) previously described [21].

### 2.5. Isolation of Nuclear Fraction from Epithelial Cells

Nuclei from primary human bronchial epithelial cells (HBEpCs) or MLE12 cells were prepared by sucrose density gradient differential centrifugation. Cells grown in 100 mm dishes to ~90% confluence were challenged with vehicle or heat-inactivated *PA* (1 × 10^8^ CFU/mL) for 2 h, washed three times with PBS, trypsinized, detached from the cell culture dishes, and centrifuged at 300× *g* for 10 min. The cell pellets were washed again with PBS and resuspended in 5 mL buffer A (10 mM HEPES-KOH/pH 7.4, 1.5 mM MgCl_2_, 10 mM KCl, and 0.5 mM dithiothreitol (DTT), and homogenized ten times using a Dounce homogenizer with a tight pestle. Cell homogenates were centrifuged at 200× *g* for 5 min at 4 °C and supernatant was collected and stored as cytoplasmic fraction. The pellet was resuspended in 3 mL of 0.25 M sucrose, 10 mM MgCl_2_, and layered over 3 mL 0.35 M sucrose containing 0.5 mM MgCl_2_, and centrifuged at 1400× *g* for 5 min at 4 °C. The nuclear pellet was collected, resuspended in radioimmunoprecipitation assay buffer (RIPA) buffer, sonicated for 10 s with a probe sonicator at a setting of 5, centrifuged at 10,000× *g* for 10 min at 4 °C, and the pellet was collected as the nuclear fraction [22]. The purity of the nuclear preparation was determined by Western blotting and immunostaining for lamin B with anti-lamin B antibody, a component of nuclear lamina.

### 2.6. Measurement of IL-6, TNF-α, IL-4, and IL-12

BAL fluids from mouse lung were centrifuged at 10,000× *g* for 10 min at 4 °C, and IL-6, TNF-α, IL-4, and IL-12 levels in the supernatants were measured using a commercial ELISA kit (R&D Systems, Minneapolis, MN, USA), according to the manufacturer’s instructions.

### 2.7. HDAC Activity

HDAC activity was measured in cell nuclei isolated from lung epithelial cells as outlined above using a commercially available kit (Cayman Chemical, Ann Arbor, MI, USA) according to the manufacturer’s instructions.

### 2.8. Measurement of H_2_O_2_

BAL fluid from mice were collected and centrifuged at 10,000× *g* for 20 min at 4 °C, and H_2_O_2_ concentration was measured using Amplex Red Hydrogen Peroxide/Peroxidase kit (Invitrogen, Carlsbad, CA, USA), according to the manufacturer’s instruction.

### 2.9. Immunoblotting and Immunoprecipitation

Immunoblotting and immunoprecipitation (IP) were performed as described previously [20,21,23]. In brief, after appropriate treatments, cells were pelleted in ice-cold PBS, lysed in cell lysis buffer (Cell Signaling, Beverly, MA, USA), and sonicated. Lysates were centrifuged at 10,000× *g* for 10 min at 4 °C, supernatants were collected, and protein was assayed using the bicinchoninic acid (BCA) protein assay kit (Thermo Fisher Scientific, Waltham, MA, USA). For immunoprecipitation (IP) experiments, equal amounts of protein (0.5–1 mg) from each sample were pre-cleared with control immunoglobulin G (IgG) conjugated to A/G agarose beads at 4 °C for 1 h, supernatants were collected and incubated overnight with primary antibody conjugated to protein A/G agarose beads at 4 °C. The next day, the samples were centrifuged at 1000× *g* for 1 min in a microcentrifuge, the beads were collected by removing supernatant buffer, and 40 µl of sodium dodecyl sulfate (SDS) sample buffer [100 mM tris(hydroxymethyl) aminomethane (Tris-HCl) (pH 6.8), 4% SDS, 0.1% bromophenol blue, 20% glycerol, 200 mM DTT) was added to the beads and boiled. Lysates were then subjected to 10% sodium dodecyl sulfate-polyacrylamide (SDS-PAGE) followed by Western blotting. Proteins were detected by immunoblotting using appropriate primary antibodies, and horse radish peroxidase (HRP)-conjugated anti-rabbit or anti-mouse secondary antibodies. Band intensities were quantified by densitometry using the NIH Image J software.

### 2.10. RAC1 Activation Assay by Western Blotting

Activation of RAC1 in response to *P. aeruginosa* was determined using the Rac1 activation kit (Cell Biolabs, Cat# STA-401-1). Activated RAC1 (guanosine triphosphate (GTP)-bound) in nuclear lysates (100–200 µg protein) from cells with and without *P. aeruginosa* treatment was pulled down with p21-binding domain (PBD) of p21-activated protein kinase (PAK), and immunodetected by Western blotting with anti-RAC1 antibody, as recommended by the manufacturer.

### 2.11. Detection of Oxidized HDAC2

HDAC2 oxidation was assayed by OxyBlot protein oxidation detection kit (EMD Millipore). Briefly, MLE12 or HBEpCs were treated with heat-inactivated *P. aeruginosa* 103 for 3 h. Cells were lysed with ice-cold lysis buffer containing 50 mM DTT for 15 min, lysates were centrifuged at 14,000× *g* for 10 min, and the supernatant was used for immunoprecipitation with HDAC2 antibody as described above. The precipitated beads were then subjected to the detection of oxidized protein as per the manufacturer’s instructions.

### 2.12. Immunohistochemical Staining

The trachea was cannulated, and the lungs were perfused with 10% neutral-buffered formalin. The left lung was removed, post fixed for 2 days at 4 °C, paraffin-embedded, and sectioned (5 μm). Sections were deparaffinized, pretreated for antigen retrieval, and incubated with antibodies against surfactant protein C (SPC) (1:50; sc-7705, Santacruz, Dallas, TX, USA) and NOX4 (1:300; 14347-1-AP, Proteintech, Rosemont, IL, USA) overnight at 4 °C, followed by respective secondary antibodies (Alexa Fluor 488 and Alexa Fluor 594 from Invitrogen at 1:500). Appropriate negative controls were run by omitting the primary antibody to confirm nonspecific staining. The immunostained sections were cover-slipped with a Vectashield mounting medium containing 4′,6-diamidino-2-phenylindole (DAPI) (Vector Laboratories, Burlingame, CA, USA) and visualized with a fluorescence microscope. Images of the immunostained sections were captured with a Rolera XR CCD camera (Q-Imaging, Surrey, BC, Canada) mounted on a microscope (Leica Microsystems, Wetzlar, Germany). Images were taken at 80× magnification and NOX4 expression (green) in SPC-stained (red) cells (yellow merged) was quantified.

### 2.13. Preparation of P. aeruginosa

*P. aeruginosa* 103 bacteria from frozen stocks, which express Type III proteins and secrete exoenzymes U and T, were streaked onto trypticase soy agar plates and grown in a deferreated dialysate of trypticase soy broth supplemented with 10 mM nitrilotriacetic acid (Sigma Aldrich, St. Luis, MO, USA), 1% glycerol, and 100 mM monosodium glutamate at 33 °C for 15 h in a shaking incubator. Cultures were centrifuged at 8500×*g* for 5 min, and the bacterial pellet was washed twice in Ringer lactate and diluted to appropriate number of colony forming units (CFU) per mL in Ringer lactate solution after measuring the optical density in a spectrophotometer. The bacterial concentration was confirmed by dilution and plating out the known dilution on sheep blood agar plates [15]. Heat inactivation was achieved by heating the suspension at 55 °C for 30 min.

### 2.14. Statistical Analysis

Results are presented as means ± standard deviations (SDs) from at least three independent experiments. The data were analyzed for statistical significance according to paired and unpaired *t* tests, and two-way analysis of variance (ANOVA) using Microsoft Excel (Microsoft Corp., Seattle, WA, USA). *p* ≤ 0.05 was considered statistically significant.

## 3. Results

### 3.1. Genetic Deletion of Nox4 in Alveolar Epithelial Cells Protects Mice from P. aeruginosa-Induced Lung Inflammatory Injury

We previously reported that the upregulation of *Nox4* in mouse lung infected with *P. aeruginosa*, and downregulation of *Nox4* with shRNA in vivo in mouse lung and in vitro in human lung endothelial cells, reduced *P. aeruginosa* mediated lung inflammatory injury and endothelial permeability [6]. While Nox4 is highly expressed in lung endothelial cells [24], its expression and role in *P. aeruginosa*-induced lung inflammation is unclear. To characterize the relative importance of lung epithelial cell NOX4 in the development of lung injury caused by *P. aeruginosa*, we generated mice with conditional knockout of Nox4 in alveolar epithelial cell (Ep) by breeding tamoxifen-inducible alveolar epithelial-specific secretory surfactant protein C (SPC)-Cre mice with *Nox4^flox/flox^* mice to generate epithelial cell-specific (*LEp-Nox4 Knock out* (*KO*); *Nox4^flox/flox^*: *SPC Nox4^Cre+^*) in C57BL background. The conditional deletion of Nox4 was achieved by administering tamoxifen to animals for 5 consecutive days prior to *P. aeruginosa* infection. Controls (*Nox4^flox/flox^*) and *Nox4 KO* (*Nox4^flox/flox^*: *SPC Nox4Cre^+^*) mice were infected with *P. aeruginosa* 103 strain (1 × 10^6^ CFU/mouse) or 20 µL sterile phosphate-buffer, and their lungs were recovered 24 h post-challenge. Intratracheal administration of *P. aeruginosa* induced lung injury as determined by the increased infiltration of peripheral blood mononuclear cells (PMNs) in lungs and BAL fluid, and increased protein levels in BAL fluid, and these responses were significantly lower in the *LEp-Nox4 KO* mice (Figure 1A–C). As reported earlier [15], *P. aeruginosa* infection enhanced ROS levels in the BAL fluid of wild-type (WT) mice, which were reduced in LEp-Nox4 KO mice (Figure 1D). Likewise, a significant elevation of pro-inflammatory cytokines IL-6 (~3-fold) and TNF-α (~6-fold) were observed in the BAL fluid at 24 h after instillation of *P. aeruginosa* in mouse lungs, which were attenuated by epithelial deletion of Nox4 (Figure 1E,F). Next, we determined if Nox4 in epithelial cells modulated the pulmonary Th2 responses. Genetic deletion of Nox4 in the alveolar epithelial cells significantly reduced IL-12 and IL-4 levels in the BAL fluid after *P. aeruginosa* infection compared to the controls (Figure 1G,H). The epithelial cell-specific knockdown of NOX4 was confirmed by immunohistochemistry of co-staining NOX4 (green) with SPC (red) in alveolar epithelial cells in paraffin-embedded sections of lungs from *Nox4^flox/flox^* and *LEp-Nox4 KO* mice (Figure 2A,B) and NOX4 immunostaining of lung tissue lysates from tamoxifen-treated control and *LEp-Nox4 KO* mice treated with *P. aeruginosa* treatment for 24 h (Figure 2C). Interestingly, *P. aeruginosa* challenge for 24 h stimulated NOX4 expression in lung tissue lysates from Nox4^flox/flox^ but not in *LEp-Nox4 KO* mice. These results show that deletion of Nox4 in the alveolar epithelial cells protected mice from *P. aeruginosa*-mediated inflammatory injury by attenuating infiltration of polymorphonuclear leukocytes (PMNs) into the lung and reducing levels of ROS and pro-inflammatory cytokines in the BAL fluid.

### 3.2. Deletion of Nox4 Reduces P. aeruginosa-Mediated H3 and H4 Histone Acetylation in Mouse Lung

Having demonstrated in vivo that alveolar epithelial NOX4 is essential for *P. aeruginosa*-mediated lung inflammatory injury, next we investigated the effect of *Nox4* knockdown on *P. aeruginosa*-mediated histone acetylation in primary alveolar epithelial cells isolated from *Nox4^flox/flox^* and *LEp-Nox4 KO* mice. Mouse alveolar Type II epithelial cells isolated from *LEp-Nox4 KO* mice showed >95% deletion of NOX4 (Figure 3A). Further, only the alveolar Type II epithelial cells isolated from *Nox4^flox/flox^* mice and not *LEp-Nox4 KO* mice showed enhanced H3K9 and H4K8 histone acetylation to *P. aeruginosa* infection (Figure 3A). Similarly, knockdown of *Nox4* with siRNA in HBEpCs reduced H3K8 and H4K9 histone acetylation (Figure 3B). Interestingly, in the alveolar epithelial cells, NOX4 is localized both in the cytoplasm and nucleus, and challenging the cells with heat-inactivated *P. aeruginosa* increased NOX4 expression in the nucleus, with a slight decrease in the cytoplasmic fraction (Figure 3C). In contrast to NOX4, no nuclear localization of NOX1, NOX2, or NOX3 was observed without or with *P. aeruginosa* treatment of HBEpCs (Figure 3C).

Earlier, we have demonstrated a role for SPHK2, but not SPHK1, in *P. aeruginosa*-mediated H3 and H4 histone acetylation in the mouse lung and HBEpCs [14]. Further, as the S1P signaling seems to modulate cellular ROS generation in lung cells, we investigated whether sphingosine kinase-regulated NOX4 expression plays a role in histone acetylation, in the mouse lung. Infection of mouse lung with *P. aeruginosa* increased NOX4 expression in WT and *Sphk1^−/−^* mice; however, there was no robust increase in the expression of NOX4 in the lungs of *Sphk2^−/−^* mice exposed to *P. aeruginosa* (Figure 3D). Together, these results show nuclear localization of NOX4 and its upregulation in the nucleus, and knockdown of *Nox4* reduced *P. aeruginosa*-mediated H3 and H4 histone acetylation in lung epithelial cells.

### 3.3. P. aeruginosa Stimulates Nuclear ROS Generation via Nox4, But Not Nox2, in Lung Epithelial Cells

Having demonstrated a role for the *LEp-Nox4* in *P. aeruginosa-*mediated lung injury and nuclear localization of NOX4 in lung epithelial cells, next we determined the role of nuclear NOX4 in the generation of hydrogen peroxide (H_2_O_2_) by probing with the fluorescent protein sensor, HyPer, that was engineered to specifically target the nucleus [25,26]. Mouse lung epithelial MLE-12 cells were transfected with HyPer prior to challenge with vehicle or heat-inactivated *P. aeruginosa* for different time periods, following which the H_2_O_2_ generation was quantified by live imaging of cells using confocal microscope. The transfection efficiency of HyPer in MLE-12 cells was ~61% as determined by the co-localization of DAPI and green fluorescence of HyPer to exogenous addition of 100 µM H_2_O_2_. As shown in Figure 4A,B, *P. aeruginosa* stimulated the nuclear H_2_O_2_ in a time-dependent manner. To further characterize the role of NOX4 and NOX2 in the enhanced H_2_O_2_ production by *P. aeruginosa*, MLE-12 cells were transfected with scrambled, *Nox2*, or *Nox4* siRNA for 48 h followed by the transfection of nuclear HyPer for 24 h prior to challenge. Downregulation of *Nox4*, but not *Nox2*, with siRNA significantly reduced the nuclear H_2_O_2_ production stimulated by *P. aeruginosa* (Figure 4C,D). Further, both *Nox2* and *Nox4* siRNA were effective in downregulating their targets (>70%) compared to the scrambled siRNA in these cells (Figure 4E). The role of NOX4 in *P. aeruginosa*-mediated H_2_O_2_ production in the nucleus was further confirmed in MLE-12 cells transfected with the nuclear HyPer and treated with NOX1/NOX4 inhibitor, Setanaxib (GKT 137831) (5 µM), which attenuated nuclear H_2_O_2_ generation mediated by the heat-inactivated *PA* challenge (Figure 5A,B). These results show NOX4, but not NOX2-dependent nuclear generation of H_2_O_2_ by *P. aeruginosa* in lung epithelial cells.

### 3.4. Downregulation of PKC δ and Sphk2 or Inhibition of SPHK2 Reduces P. aeruginosa-Induced Nuclear ROS in Lung Epithelial Cells

We have shown earlier that activation of *PKC δ* by *P. aeruginosa* induced SPHK2 phosphorylation and translocation to the nucleus with increased S1P generation, H3/H4 histone acetylation, and IL-6 secretion in the lung epithelial cells [14]. As *P. aeruginosa* challenge stimulated the nuclear H_2_O_2_ production via nuclear NOX4, we determined the role of PKC *δ* and SPHK2 in nuclear H_2_O_2_ generation. Knockdown of *Pkc δ* or *Sphk2* with siRNA in MLE12 epithelial cells transfected with the nuclear HyPer significantly reduced nuclear H_2_O_2_ production stimulated by *P. aeruginosa* (Figure 6A–D). Similarly, inhibition of SPHK2 with ABC294640 [14,27] also attenuated the nuclear H_2_O_2_ generation (>80%) compared to cells not treated with the SPHK2 inhibitor (Figure 6E,F). These results show the critical requirement of PKC *δ* and SPHK2 in the NOX4-dependent nuclear H_2_O_2_ production in lung epithelial cells.

### 3.5. P. aeruginosa-Induced Oxidation of Nuclear HDAC2 Is Dependent on PKC δ, SPHK2, and NOX4 Signaling in Lung Bronchial Epithelial Cells

Having demonstrated a role for PKC *δ* and SPHK2 in the nuclear NOX4-induced H_2_O_2_ production, next we investigated whether nuclear H_2_O_2_ generated by *P. aeruginosa* inhibited HDAC activity and enhanced protein oxidation of HDAC2. Primary HBEpCs were challenged with heat-inactivated *P. aeruginosa* for 1–3 h and HDAC activity was determined in the nuclear fraction. As shown in Figure 7A, *P. aeruginosa* exposure to HBEpCs inhibited HDAC1/2 activity in the nuclear fraction, in a time-dependent manner. To determine if nuclear H_2_O_2_ facilitated oxidation of HDAC1/2, cells were transfected with scrambled, *Pkc δ, SphK2, Nox4*, or *Nox2* siRNA (100 nM, 48 h). The cells were then exposed to *P. aeruginosa* for 3 h, and cell lysates were immunoprecipitated with anti-HDAC1/2 antibody and oxidation of carbonyl groups was detected by Western blotting with the OxyBlot protein oxidation detection kit. As shown in Figure 7B–E, *P. aeruginosa* exposure enhanced the oxidation of carbonyl groups in HDAC2, which was attenuated in cells wherein *Nox4*, *PKC δ*, and *Sphk2* were downregulated by specific siRNA, but not *Nox2* siRNA. Similarly, pretreatment of cells with *N*-acetyl-L-cysteine, an antioxidant, reduced the *P. aeruginosa*-mediated HDAC2 oxidation (Figure 7F–G). These results demonstrate that the nuclear H_2_O_2_ enhances HDAC2 protein oxidation that is dependent on PKC *δ*, SPHK2, and NOX4 in lung bronchial epithelial cells.

### 3.6. Inhibition of RAC1 Reduces P. aeruginosa-Mediated Nuclear ROS Production

RAC1 is an essential cofactor required for NOX1-, NOX2-, and NOX3-mediated superoxide generation in mammalian cells [28,29,30]; however, there seems to be no requirement of RAC1 in NOX4-dependent ROS production under the basal or stimulated conditions [31]. To determine the role of RAC1 in nuclear ROS production, HBEpCs were transfected with the nuclear-HyPer prior to *P. aeruginosa* challenge. As shown in Figure 8A,B, treatment with RAC1-guanine nucleotide-exchange factor (GEF) inhibitor NSC23766 attenuated the *P. aeruginosa-*mediated nuclear HyPer fluorescence. Interestingly, RAC1 was immunodetected in the nuclear fractions of lung epithelial cells, and *P. aeruginosa* stimulated RAC1 activation, as determined by the p21-activated kinase 1 (PAK1) pulldown assay, which was blocked by a catalytically inactive dominant-negative PKC δ mutant (Figure 8C,D). These results suggest a role for RAC1 in nuclear H_2_O_2_ production stimulated by *P. aeruginosa* in the lung epithelial cells.

### 3.7. P. aeruginosa Enhances Association of NOX4, PKC δ, and RAC1 with HDAC2 in Nucleus of Lung Epithelial Cells

Having shown a potential role for RAC1 in nuclear ROS production by *P. aeruginosa*, next we investigated the potential association between NOX4, PKC δ, and RAC1 with HDAC1/2 in the nucleus. HBEpCs were treated with the heat-inactivated *P. aeruginosa* for 3 h, following which the cell lysates were immunoprecipitated with the anti-HDAC2 antibody and immunoblotted. As shown in Figure 8E,F, *P. aeruginosa* enhanced the association of HDAC2 with NOX4, p-PKC δ, and RAC1, suggesting a potential role for this complex in the nuclear ROS generation.

## 4. Discussion

ROS, including superoxide, hydroxyl radical, nitric oxide, peroxynitrite, hydrogen peroxide, and others, are generated mostly in the mitochondria, peroxisomes, endoplasmic reticulum, lamellipodia, as well as in the cytoplasm of aerobic cells [23,32,33]. In addition, aging and several external agents such as radiation, environmental toxins, allergens, particulate matter, bacteria, and viruses can also trigger ROS production, especially in the lung. While low levels of ROS generated under normal physiological condition regulate cellular homeostasis and host defense, increased ROS accumulation due to impaired antioxidant defense system can induce damage of cellular proteins, lipids, and DNA [34,35]. Also, there is evidence for increased ROS levels to activate stress signals in cells that may have either detrimental or protective function [36]. Increased ROS production has been implicated in several human pathologies, including atherosclerosis, diabetes, respiratory diseases, and lung cancer [37,38,39]. *P. aeruginosa* is a common Gram-negative bacterium associated with respiratory tract infection in immunocompromised patients in diverse clinical settings, including cystic fibrosis [3,4]. *P. aeruginosa* infection induces generation of ROS as an innate immune response; however, uncontrolled ROS accumulation results in endothelial and epithelial barrier dysfunction and lung damage [10,14,15,40]. Several lung cell types could be involved in *P. aeruginosa-*mediated ROS generation; however, involvement of specific cell types and subcellular organelle(s) in this process is unclear. Here, using cell-specific knockdown of *Nox4* in mouse lung, we have identified a novel role for lung alveolar epithelial NOX4 in ROS generation, histone acetylation, and inflammation induced by *P. aeruginosa*. Further, in vitro results with lung epithelial cells show, for the first time, a role for nuclear NOX4, but not NOX2, generating ROS in the nucleus, which regulates HDAC activity, and H3 and H4 histone acetylation. Further, nuclear NOX4 generated by *P. aeruginosa* induced HDAC1/2 oxidation, which was blocked by knockdown of PKC δ, Sphk2, and Nox4 in lung epithelial cells. Thus, our results establish a link between NOX4 activation and generation of nuclear ROS that modulates HDAC activity and histone acetylation via oxidation of HDAC1/2.

The NOX family, comprising of the family members NOX1–5, are differentially expressed in mammalian cells [29,41]. NOX enzymes are involved in the development of several lung pathologies. Lung epithelial cells express NOX2 and NOX4, and NOX4 is crucial for epithelial apoptosis and fibroblast to myofibroblast differentiation in bleomycin-induced lung fibrosis in mice [42]. Lung infection by *Pseudomonas* and other bacteria induces oxidative stress via NOX proteins [15,16,40]; however, the role of the alveolar epithelium is unclear. In this study, we have demonstrated a critical role for alveolar epithelial NOX4 in promoting oxidative stress and lung injury by *P. aeruginosa* in mouse lung as epithelial specific knockdown of *Nox4* offered protection against pulmonary inflammation and injury. Deletion of *Nox4* in the alveolar epithelial cells reduced ROS levels in the BAL fluid compared to the control mice; however, the ROS levels were still higher compared to the non-*PA* challenged mice (Figure 1D). A plausible explanation for this may be due to the contribution from lung endothelial cells that express both NOX2 and NOX4 [24]. Interestingly, NOX4 seems to have an important modulatory role in histone acetylation, as deletion of *Nox4* reduced the *P. aeruginosa*-mediated H3K8 and H4K9 acetylation profile in lung tissue lysates and alveolar type II epithelial cells isolated from *LEp-Nox4 KO* mice (Figure 3).

Our in vitro studies with lung epithelial cells establish, for the first-time, a novel role for NOX4 in *P. aeruginosa*-mediated nuclear ROS production and it’s signaling. In HBEpCs, NOX4 was localized in both the cytoplasm and nucleus, and *P. aeruginosa* increased nuclear expression of NOX4 3 h post-challenge. A similar increase in nuclear NOX4 expression to stimulation by phenylephrine was observed in cardiac myocytes [18]. Interestingly, in lung epithelial cells, *P. aeruginosa* enhanced nuclear NOX4 expression after 3 h of post-challenge, while in mesangial cells and cardiac myocytes, NOX4 expression was upregulated within minutes of angiotensin II or phenylephrine treatment [18,43]. In both cardiac myocytes [18] and lung endothelial cells [15], the increased NOX4 expression was regulated by NF-kB; however, it is unclear if NOX4 activity was also modulated by these agents. In contrast to NOX2 regulation by p47*^phox^*, p67*^phox^*, p22*^phox^*, and RAC1 [28,29,30,31], NOX4 activity is not dependent on cytosolic cofactors [33]. However, recent evidence suggests that polymerase (DNA-directed) delta-interacting protein 2 modifies the activity and/or function of NOX4 in smooth-muscle cells [44], while NRF2 regulates hyperoxia-induced NOX4 via ARE binding elements in lung endothelial cells [45]. Another important finding here is a potential regulation of NOX4 expression by SPHK2 in mouse lung. Deletion of *Sphk2*, but not *Sphk1*, modulated *P. aeruginosa-*stimulated NOX4 expression in mouse lung (Figure 3D), suggesting that S1P generated in the nucleus and not in the cytoplasm might be involved in regulating NOX4 expression. However, deletion of *Sphk2* in the mouse had no significant effect on *Nox4* mRNA level [8], suggesting the degradation of NOX4 possibly via the ubiquitination pathway, which needs further investigation.

The current study highlights a unique role for nuclear NOX4 generated ROS that induces chromatin structure as well as chromatin reader and repair properties via hyper-acetylation. Histones are the most common chromatin proteins, and modification of histones by ROS changes their abundance, structure, or post-translational modifications that can impact gene expression, genome stability, and replication [46,47]. As revealed in the current study, deletion of *LEp-Nox4* significantly reduced *P. aeruginosa*-induced Th1 and Th2 cytokines in BALF compared to the control group (Figure 1), suggesting potential involvement of the NOX4-derived ROS in modulation of promoter regions of pro-inflammatory genes through the acetylation of lysine residues of H3 and H4 histones. Indeed, in this study, we demonstrated a role for SPHK2/S1P signaling in increased epigenetic marks H3 lysine9 (H3K9ac) at nuclear factor kappa B binding sites on *IL6* promoter after *P. aeruginosa* challenge of lung epithelial cells, which was blocked by SPHK2 inhibitor ABC294640 and not by SPHK1 inhibitor PF543 [14]. Acetylation of H3K9 and H4K16 have been defined to play a key role in causing relaxation of chromatin structure at transcriptional level of genes [46,48,49]. While *P. aeruginosa* induced hyperacetylation of H3K9 and H4K8 in lung epithelium by inhibiting HDAC1/2 activity, Ni^2+^ caused histone hypoacetylation via increased ROS in hepatoma cells through decreased histone acetyl transferase (HAT) activity [50], indicating differential role of ROS in modulating HAT and/or HDAC activity.

Emerging evidence suggests that ROS can modulate HDACs through S-glutathionylation, S-nitrosylation, phosphorylation, and oxidation, leading to increased or decreased histone acetylation and restructuring of the chromatin [51]. A novel finding of this study is highlighted by the inhibition of HDAC1/2 activity caused by the nuclear NOX4-derived ROS that was stimulated by *P. aeruginosa*, and thereby enhancing histone acetylation in the lung epithelial cells. We also observed that *N*-acetylcysteine caused the inhibition or downregulation of PKC *δ*, SPHK2, and NOX4 attenuated oxidation of HDAC1/2 and *P. aeruginosa-*induced H3/H4 histone acetylation, indicating a plausible link between HDAC1/2 oxidation, HDAC activity, and chromatin restructuring in the epithelium. In contrast to the epithelial nuclear HDAC1/2 oxidation, NOX4-dependent oxidation promoted nuclear exit of HDAC4 in mouse heart, while mice with NOX4 deficiency in cardiac myocytes were protected against pressure-overload-induced cardiac hypertrophy [18]. However, in another study, an opposing effect of NOX4 on cardiac dysfunction was observed. Deletion of endogenous NOX4 (NOX4 null mice) exaggerated the load-induced cardiac dysfunction in mice, while cardiomyocyte-targeted NOX4 expression protected the mice against load-induced cardiac dysfunction by enhancing angiogenesis [52]. This discrepancy on NOX4-mediated exaggeration or protection of cardiac dysfunction is unclear but may be due to differential distribution of NOX4 and generation of ROS in a different cellular location. The functional consequence of HDAC1/2 modulation by ROS is varied in different cell types. In lung epithelium, *P. aeruginosa*-mediated oxidative stress causes hyperacetylation of H3/H4 histones, which is an important factor involved in opening the chromatin for the development of lung inflammatory injury. There is evidence for the increased oxidative stress and peroxynitrite formation in COPD and severe asthma, and reduced HDAC2 activity due to HDAC2 tyrosine nitration, and enhanced IL-8 gene expression in bronchial epithelial cells [53,54]. Further, depending on the cell type(s), oxidative stress can also lead to hyper- or hypo-phosphorylation of HDACs. In human umbilical vein and microvascular endothelial cells, increased HDAC4 phosphorylation by NOX4-derived ROS is essential to maintain proper endothelial tube formation [55]. However, reversal of NOX4-mediated oxidation of HDAC4 by reductase Trx1 inhibited its nuclear export independent of its phosphorylation status [56].

Another salient feature of this study is the potential interaction between HDAC2, NOX4, p-PKC δ, and RAC1 under basal condition and an enhancement in the association after *P. aeruginosa* challenge of the lung epithelial cells. It is intriguing that the interaction between HDAC1/2, NOX4, and p-PKC *δ* is critical in the spatiotemporal generation of nuclear ROS generation and inhibition of HDAC1/2 activity. HDAC1 and HDAC2 are core components of the Sin3 and nucleosome remodeling and deacetylase (NuRD) repressor complexes that regulate chromatin structure [57]. We have previously shown SPHK2 to be an integral component of the nuclear HDAC repressor complex Sin3/NuRD and blocking SPHK2 or PKC *δ* attenuated *P. aeruginosa-*mediated interaction between HDAC1/2 and SPHK2 and H3 and H4 histone acetylation [14]. This interaction between the Sin3/NuRD repressor complex and the PKC *δ*/SPHK2/S1P signaling seems to be essential for the spatiotemporal generation of S1P in the nucleus. A previous study in the Michigan Cancer Foundation-7 (MCF-7) breast cancer cell line showed phorbol myristate acetate-induced translocation of SPHK2 to the nucleus and generation of nuclear S1P, which inhibited HDAC activity [58]. *P. aeruginosa*, in addition to NOX4 pathway, also stimulates nuclear S1P generation and histoneH3/H4 acetylation that is SPHK2-dependent and regulated by HDAC1/2 [14,59], indicating at least two potential inter-connected or independent pathways in regulating HDAC1/2 activity and histone acetylation. Genetic deletion of *Sphk2* in mouse reduces *P. aeruginosa-*mediated NOX4 expression in mouse lung (Figure 3); however, it is unclear how SPHK2/S1P signaling regulates NOX4 in lungs. Our intriguing results show the RAC1 involvement in *P. aeruginosa*-induced nuclear ROS generation mediated by NOX4 (Figure 8); however, these data do not implicate a direct role of Rac1 in regulating NOX4 activity in the nucleus of lung epithelial cells. In an earlier study, an indirect role of RAC1 in NOX4/ROS-dependent activation of AKT/protein kinase B pathway in mesangial cells was reported [60]. *P. aeruginosa*-mediated activation of PKC *δ* stimulated SPHK2 phosphorylation and its translocation to the nucleus in lung epithelial cells [14], while our current study showed RAC1 to be downstream of PKC *δ* in the nuclear signaling cascade; however, it is unclear if RAC1 activation is upstream or downstream of the SPHK2/S1P pathway. It has been shown in vitro that NOX4 activity does not require the cytosolic components such as RAC1, *p47^phox^*, and p67*^phox^* [31], although p22*^phox^* is an integral component of NOX4 [24,61,62]. Further studies are necessary to delineate the role of RAC1 in the activation of NOX4/ p22*^phox^* and other interacting proteins in this context.

## 5. Conclusions

There is overwhelming evidence for intracellular ROS to play a pivotal role in signal transduction under normal physiology and induce oxidative stress in various human pathologies. Among the various sources of ROS, the NOX1–5 family has been shown to be involved in a wide variety of physiological and pathophysiological processes. While NOX1, 2, 3, and 5 generate superoxide, NOX4 produces predominantly hydrogen peroxide (H_2_O_2_), and is localized both in the cytosolic compartment and nucleus of the lung epithelial cell. Our findings collectively show a novel role for *PA*-induced PKC δ→SPHK2→RAC1→NOX4 activation and nuclear ROS generation. Further, the nuclear ROS modulates H3 and H4 histone acetylation and chromatin remodeling by inhibiting HDAC1/2 activity via oxidation and regulates secretion of pro-inflammatory cytokines in the lung epithelium (Figure 9). Thus, NOX4 could be a potential therapeutic target in mitigating bacterial and other lung inflammatory injury.

## Figures and Tables

**Figure 1 antioxidants-10-00477-f001:**
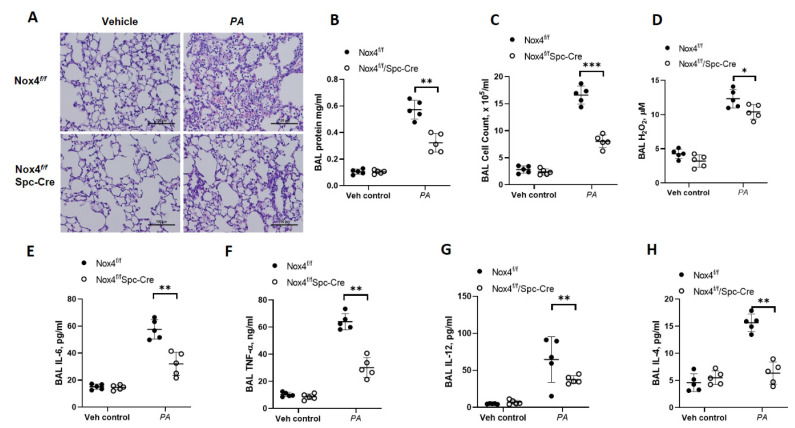
Deletion of NADPH Oxidase 4 (NOX4) in alveolar type II cells of mice reduced *P. aeruginosa*-induced lung inflammatory injury. Nox4*^flox/flox^*and LEp-Nox4 knock out (KO) mice were challenged intratracheally with sterile PBS or *P. aeruginosa* (*PA*) (1 × 10^6^ CFU/mouse) for 24 h. Lungs were lavaged, bronchoalveolar lavage (BAL) fluid collected, formalin fixed, embedded in paraffin, and cut into 5 μm sections for Hematoxylin-Eosin (H&E) staining. (**A**) Representative H&E staining photomicrograph of lung section from *Nox4^flox/flox^*, LEp-Nox4 KO mice with/without *PA* challenge, original magnification ×20. Scale bar = 100 μm. (**B**–**H**) BAL fluids from control and *PA*-challenged *Nox4^flox/flox^*, *LEp-Nox4* KO mice were analyzed for protein (**B**), infiltrating cells (**C**), Hydrogen peroxide (H_2_O_2_) levels (**D**), and pro-inflammatory cytokines (**E**–**H**). Data are expressed as means ± SD from one experiment (number of animals per group = 5). * *p* < 0.05, ** *p* < 0.01, *** *p* < 0.005.

**Figure 2 antioxidants-10-00477-f002:**
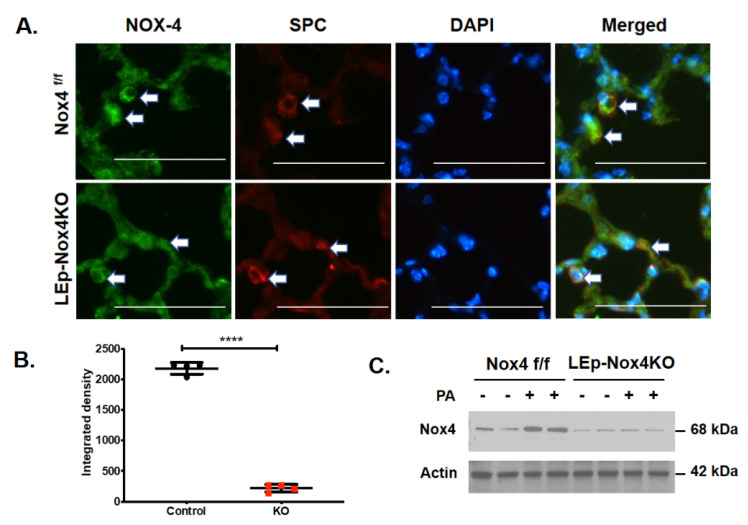
NADPH Oxidase 4 (NOX4) expression is downregulated in lungs and isolated alveolar epithelial cells from *LEp-Nox4* knock out (*KO*) mice. (**A**) Lungs from tamoxifen-treated *Nox4^flox/flox^* and *LEp-Nox4* knock out (*KO*) mice were removed, fixed, and embedded in paraffin, as described under experimental procedures. Cut 5 µm lung sections were subjected to immunofluorescence staining for NAPH Oxidase 4 (NOX4) (green), surfactant protein C (SPC) (red), and nuclear staining with 4′,6-diamidino-2-phenylindole (DAPI) (blue). Shown is a representative micrograph of co-localization of NOX4 and SPC (merge of green and red) with the alveolar type 2 pneumocytes indicated with arrows. Co-localization of NOX4 (green) and SPC (red) seen in lungs of *Nox4^flox/flox^* mice, but not in *LEp-Nox4*-deficient mice. The white triangular arrows highlight the NOX4-SPC colocalized areas in the image. Shown is a representative micrograph, Scale bar, 100 µm. (**B**) Quantification of merged images of NOX4 (green) and SPC (red) of the micrographs from (**A**). **** *p* < 0.005. (**C**) *Nox4^flox/flox^* and *LEp-Nox4* KO mice (*n* = 5) were challenged intratracheally with sterile PBS or *P. aeruginosa* (*PA*) (1 × 10^6^ CFU/mouse) for 24 h as indicated in Figure 1. Lungs were lavaged, and lung tissue lysates (30 µg protein) were subjected to sodium dodecyl sulfate-polyacrylamide (SDS-PAGE) and Western blotting, and immunostained for NOX4 and actin using specific antibodies. Shown is a representative blot from three independent experiments.

**Figure 3 antioxidants-10-00477-f003:**
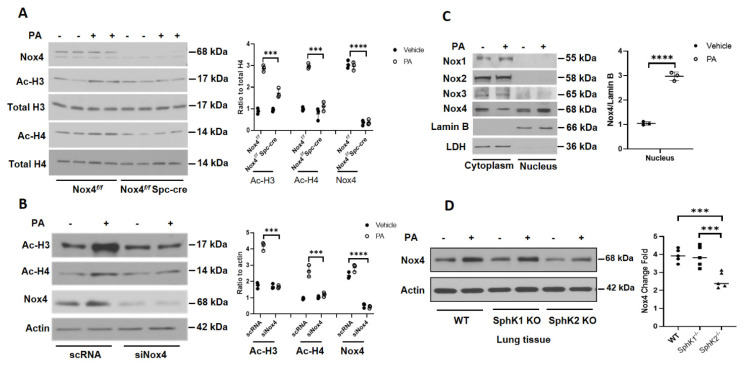
Deletion of *Nox4* reduces *P. aeruginosa-*induced histone acetylation in primary alveolar type II cells isolated from *Nox4^flox/flox^* and *LEp-Nox4 knock out* (*KO*) mice. (**A**) Primary alveolar type II cells were isolated from *Nox4^flox/flox^* and *LEp-Nox4 KO* mice and challenged with heat-inactivated *P. aeruginosa* (*PA*) (1 × 10^8^ CFU/mL) for 3 h, as described under Experimental Procedures. Cell lysates were subjected to SDS-PAGE and Western blotting and probed for NADPH Oxidase 4 (NOX4), acetylation of histone proteins H3 and H4, with specific antibodies to H3K9, H4K8, and total H3 and H4 histone antibodies. Shown is a representative blot and values are means ± SD from three independent experiments. *** *p* < 0.005, **** *p* < 0.001 between *PA* groups. (**B**) Primary human bronchial epithelial cells (HBEpCs) were transfected with scRNA or *Nox4* siRNA (50 nM) for 48 h before heat-inactivated *PA* (1 × 10^8^ CFU/mL) challenge for 3 h. Acetylation of histone proteins H3 and H4 and knockdown of *Nox4* were evaluated by Western blotting. Shown is a representative blot and values are means ± SD of three independent experiments. *** *p* < 0.005, **** *p* < 0.001 between *PA* groups. (**C**) Primary HBEpCs were treated with vehicle or heat-inactivated *PA* (1 × 10^8^ CFU/mL) for 3 h and then trypsinized. Cytoplasm and nuclear fractions were isolated by sucrose density gradient centrifugation as described in Experimental Procedures. Expression of NOX1-4 proteins was detected by Western blotting. Lamin B and LDH were used as markers for nucleus and cytoplasm, respectively. **** *p* < 0.001 compared with vehicle. (**D**) Wild type, *Sphk1^−/−^*, and *Sphk2^−/−^* mice (number of mice per group = 5) were challenged intratracheally with sterile PBS or *PA* (1 × 10^6^ CFU/mouse) for 24 h. Lung tissue lysates (30 µg protein) were subjected to SDS-PAGE and Western blotting for NOX4 and actin using specific antibodies. Shown is a representative blot and values are means ± SD from three independent experiments. *** *p* < 0.005.

**Figure 4 antioxidants-10-00477-f004:**
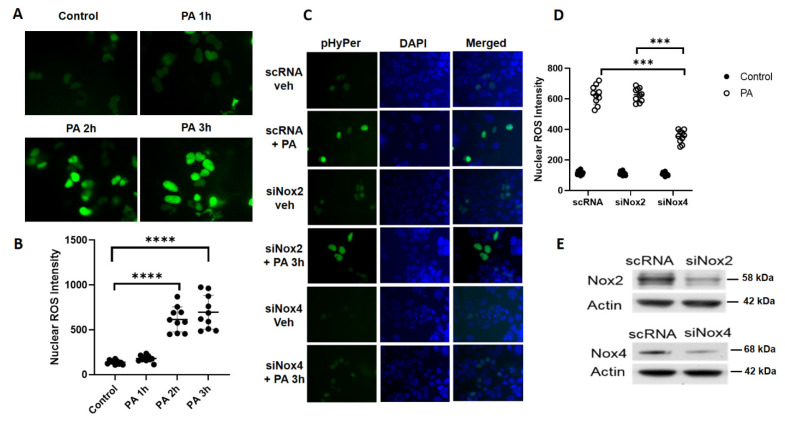
NOX4, but not NOX2, mediates nuclear ROS generation in lung epithelial cells challenged by *P. aeruginosa*. (**A**) Mouse lung epithelial (MLE) 12 cells grown in 35 mm glass-bottom dishes (~90% confluence) were transfected with scrambled or nuclear reactive oxygen species (ROS) biosensor nuclear-HyPer plasmid (50 nM) for 48 h before exposure to heat-inactivated *P. aeruginosa* (*PA*) (1 × 10^8^ CFU/mL). Nuclear ROS (hydrogen peroxide) production was monitored by live imaging of cells at varying time periods using confocal microscope. Shown is a representative micrograph depicting the green biosensor after *PA* treatment from three independent experiments. (**B**) Nuclear ROS from (**A**) were quantified by Image J and presented as intensity (at least 20 cells were chosen from each field and minimum of 3–5 different fields from each dish for analysis). Value are means ± SD from three independent experiments. **** *p* < 0.001. (**C**) MLE12 cells grown to ~70% confluence in 35 mm glass-bottom dishes were transfected with scrambled, *Nox2*, or *Nox4* siRNA (50 nM) for 48 h followed by transfection with nuclear-Hyper plasmid as indicated in (**A**) for 24 h. Cells were challenged with heat-inactivated *PA* (1 × 10^8^ CFU/mL) for 3 h, and production of ROS (hydrogen peroxide) in nucleus was monitored by live imaging (ROS, green; nucleus, DAPI) using a confocal microscope. (**D**) Nuclear ROS production was quantified by Image J and presented as intensity, at least 20 cells from each field and 3–5 fields from each dish were analyzed. Value are means ± SD from three independent experiments. *** *p* < 0.005. (**E**) Cell lysates (20 µg protein) from control, *Nox2*, and *Nox4* siRNA-treated cells were subjected to Western blotting to confirm knock down of *Nox2* and *Nox4* after treatment with specific siRNA compared to scrambled siRNA treated cells. Representative blot is shown.

**Figure 5 antioxidants-10-00477-f005:**
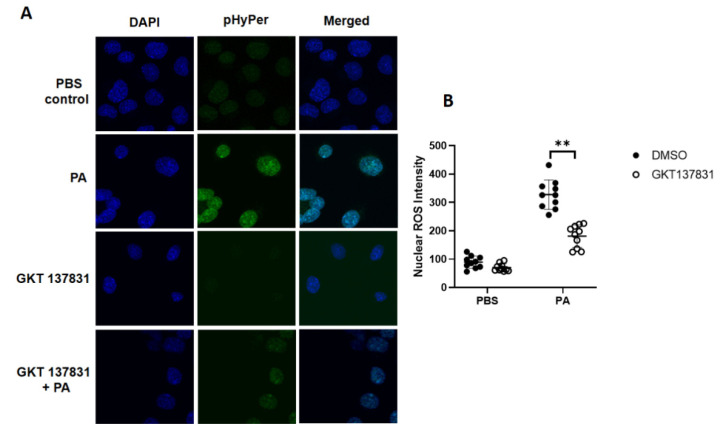
NOX4 inhibitor, Setanaxib (GKT 137831), attenuates *P. aeruginosa*-induced nuclear ROS in lung epithelial cells. Mouse lung epithelial (MLE12) cells grown in 35 mm glass-bottom dishes (~90% confluence) were transfected with control or nuclear-HyPer plasmid (50 nM) for 24 h. Cells were pretreated with NOX4/NOX1 inhibitor, Setanaxib (GKT 137831) (5 µM), for 1 h followed by heat-inactivated *P. aeruginosa* (PA) (1 × 10^8^ CFU/mL) challenge for 3 h. Nuclear ROS (hydrogen peroxide) production was detected by live imaging of cells using a confocal microscope (**A**) and quantified by NIH Image J software (**B**) Shown is a representative image from three independent experiments, and values are means ± SD from three independent experiments. ** *p* < 0.001.

**Figure 6 antioxidants-10-00477-f006:**
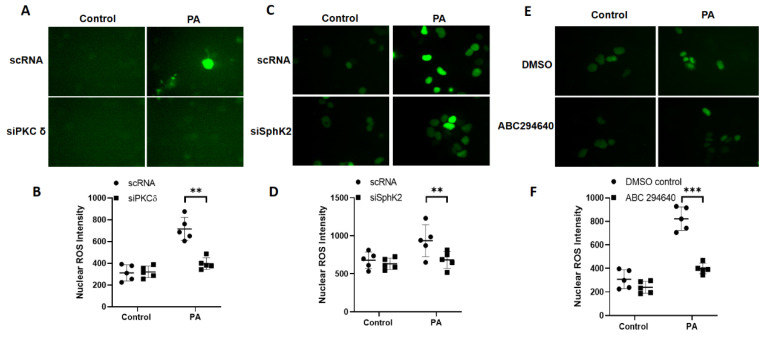
Downregulation of *Pkc δ* and *Sphk2* or inhibition of SPHK2 activity reduces *P. aeruginosa*-induced nuclear ROS in lung epithelial cells. (**A**–**D**) Mouse lung epithelial (MLE12) cells grown in 35 mm glass-bottom dishes (~70% confluence) were transfected with scrambled, *Pkc δ*, or *Sphk2 siRNA* (50 nM) for 48 h followed by transfection with nuclear ROS biosensor nuclear-HyPer plasmid (50 nM) for 24 h. Cells were treated with heat-inactivated *P. aeruginosa* (*PA*) (1 × 10^8^ CFU/mL) for 3 h, and nuclear ROS (hydrogen peroxide) production was monitored by live imaging of cells using a confocal microscope. Shown are representative micrographs for scrambled and *Pkc δ* (**A**,**B**) and *Sphk2* (**C**,**D**) siRNA-treated cells wherein nuclear ROS intensities were quantified by Image J. Values are means ± SD from three independent experiments. ** *p* < 0.01. (**E**,**F**) MLE12 cells were pretreated with sphingosine kinase 2 (SPHK2)-specific inhibitor, ABC294640 (10 µM), for 1 h prior to challenge with heat-inactivated *PA* (1 × 10^8^ CFU/mL) for 3 h, and nuclear ROS (hydrogen peroxide) production was monitored by live imaging of cells using a confocal microscope. Shown is a representative micrograph. Values are means ± SD from three independent experiments. *** *p* < 0.001.

**Figure 7 antioxidants-10-00477-f007:**
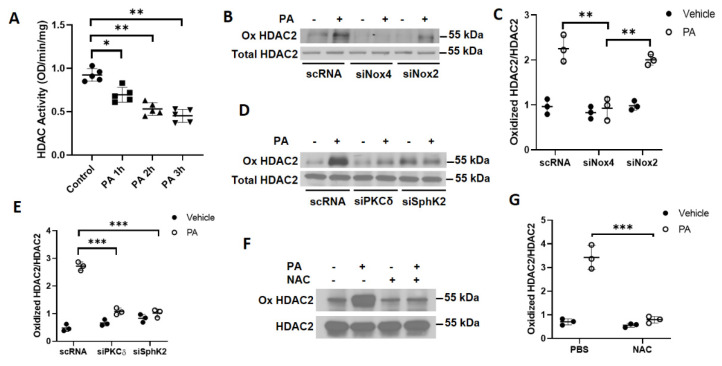
*P. aeruginosa* inhibits HDAC2 activity via PKC δ- and NOX4-dependent oxidation of HDAC2. (**A**) Primary human bronchial epithelial cells (HBEpCs) were treated with heat-inactivated *PA* (1 × 10^8^ CFU/mL) for 1, 2, and 3 h, and nuclei were isolated as described under Methods. Total HDAC activity in the nuclear fraction was determined and value expressed as means ± SD of three independent experiments. ** *p* < 0.01 * *p* < 0.05. (**B**,**C**) HBEpCs cells were transfected with *scRNA*, *siNox2*, or *siNox4* (50 nM) for 48 h prior to exposure to heat-inactivated *PA* (1 × 10^8^ CFU/mL) for 2 h. Cells were trypsinized, and nuclei were isolated as described in Experimental Procedure. Nuclear lysates (~500 µg protein) were subjected to immunoprecipitation with anti-histone deacetylase (HDAC) 2 antibody and oxidized HDAC2 (Ox HDAC2) was detected in the immunoprecipitates by Western blotting by probing with the OxyBlot oxidized protein detection kit. Shown is a representative blot, and quantification of the band intensity was carried out by the Image J program. Values are means ± SD of three independent experiments. ** *p* < 0.01. (**D**,**E**) HBEpCs cells were transfected with scrambled, *PKC δ*, or *Sphk2* siRNA (50 nM) for 48 h prior to exposure to heat-inactivated *PA*, as described above. Lysates (~500 µg protein) from isolated nuclei were subjected to immunoprecipitation with anti-HDAC2 antibody and oxidized HDAC2 was detected. Shown is a representative blot, and quantification of the band intensity was carried out by the Image J program and normalized to total HDAC2. Values are means ± SD of three independent experiments. *** *p* < 0.005. (**F**,**G**) HBEpCs cells were pretreated *N*-acetyl cysteine (NAC) (1 mM) for 30 min prior to *PA* (1 × 10^8^ CFU/mL) challenge for 2 h. Nuclei were isolated and oxidized HDAC2 (Ox HDAC2) was detected by Western blotting as described above. Values are means ± SD of three independent experiments. *** *p* < 0.005.

**Figure 8 antioxidants-10-00477-f008:**
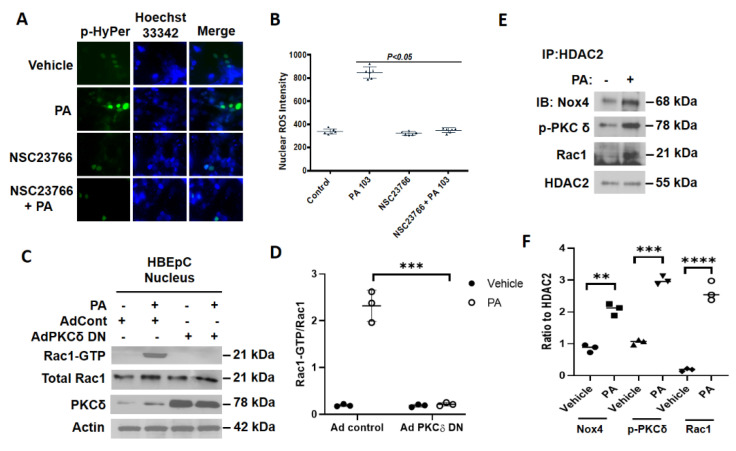
Role of RAC1 in *P. aeruginosa*-mediated nuclear ROS production and potential interaction between RAC1, PKC δ, NOX4, and HDAC2 in lung epithelial cells. (**A**,**B**) Primary human bronchial epithelial cells (HBEpCs) cells were transfected with nuclear-HyPer plasmid (50 nM) for 48 h, cells were pretreated with RAC1 inhibitor NSC23766 (5 µM) for 1 h, followed by heat-inactivated *P. aeruginosa* (*PA*) (1 × 10^8^ CFU/mL) challenge for 3 h. Nuclei were stained in situ with Hoechst 33342. Nuclear ROS (hydrogen peroxide) was imaged by confocal microscopy and quantified by Image J. Value are means ± SD from three independent experiments. *p* < 0.05. (**C**,**D**) HBEpCs were infected with vector control or PKC δ-dominant-negative adenovirus (10 MOI) for 48 h prior to exposure to heat-inactivated *PA* (1 × 10^8^ CFU/mL) for 3 h. Nuclei were isolated from vehicle and *PA*-treated cells as described in Experimental Procedure. The activated RAC1 formed was pulled down from lysates with the p21-activated kinase 1(PAK)-p21-binding domain (PBD) affinity kit described under Methods. Shown is a representative Western blot depicting nuclear RAC1 activation by *PA*, which was blocked by dominant-negative PKC δ. Total RAC1 and actin served as loading controls. Quantification of activated RAC1 was performed by the Image J program. Values are means ± SD from three independent experiments. *** *p* < 0.005. (**E**,**F**) HBEpCs were challenged with vehicle or *PA* (1 × 10^8^ CFU/mL) for 3 h. Nuclei were isolated as described in Experimental Procedure and nuclear lysates (~500 µg protein) were subjected to immunoprecipitation with anti-HDAC2 antibody. The immunoprecipitates were subjected to SDS-PAGE and Western blot analysis with anti-HDAC2, anti-NOX4, anti-RAC1, and anti-PKC δ antibodies to detect interaction of HDAC2 with NOX4, PKC δ, and RAC1. Shown is a representative blot, and values are means ± SD from three independent experiments. ** *p* < 0.01, *** *p* < 0.005, **** *p* < 0.001.

**Figure 9 antioxidants-10-00477-f009:**
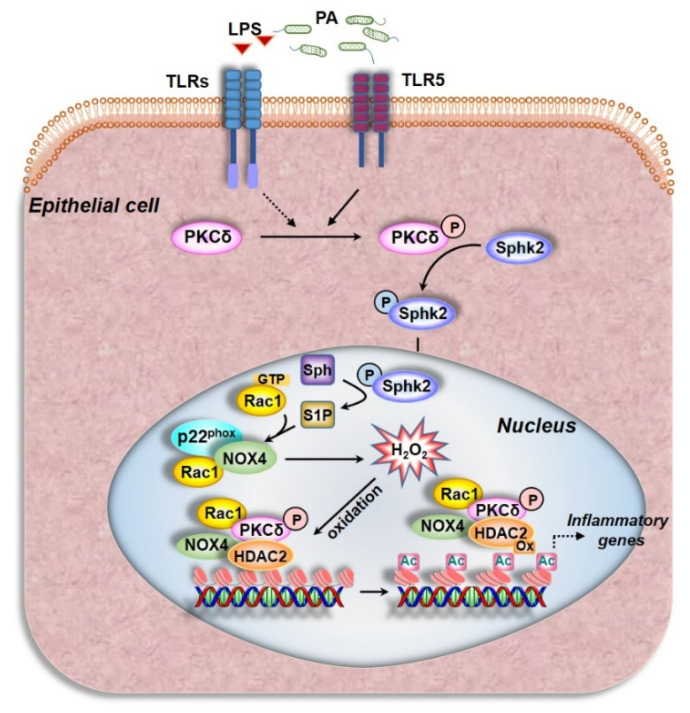
Schema depicting the role of PKC δ, RAC1, and SPHK2 signaling in nuclear NOX4-dependent ROS generation, HDAC2 oxidation, and H3/H4 histone acetylation mediated by *P. aeruginosa* in the lung epithelium. *P. aeruginosa* (*PA*) infection of lung epithelial cells in mouse lung or in culture phosphorylates and activates PKC δ, via Toll-like receptor (TLR) 5, which phosphorylates SPHK2 generating nuclear S1P. *P. aeruginosa* via lipopolysaccharide (LPS) and TLR2/4 might stimulate and phosphorylate PKC δ; however, LPS does not increase nuclear ROS mediated by *PA*. The PKC δ/SPHK2 signaling increases nuclear ROS (hydrogen peroxide) via nuclear NOX4 as downregulation or inhibition of NOX4 reduces nuclear ROS. *PA* also activates nuclear RAC1 that is PKC δ-dependent. Inhibition of RAC1 with an inhibitor, NSC23766, reduces *PA*-induced nuclear ROS in lung epithelial cells. Generation of NOX4-dependent nuclear ROS modulates HDAC1/2 activity by oxidation of HDAC1/2. Further, in the nucleus, *PA* enhances the association between HDAC1/2 and NOX4, PKC δ, and RAC1. Thus, nuclear NOX4 modulation of HDAC1/2 activity via its oxidation leads to chromatin remodeling, which can potentially regulate inflammation. The dotted lines in the schema indicate pathways yet to be demonstrated, in vitro, in lung epithelial cells. *PA*, *Pseudomonas aeruginosa*; PKC, protein kinase C; SPHK2, sphingosine kinase 2; S1P, sphingosine-1-phosphate; Sph, sphingosine; NOX4, NADPH Oxidase 4; Ox HDAC, oxidized histone deacetylase; LPS, lipopolysaccharide; Rac1, Ras-related C3 botulinum toxin substrate 1; TLR, toll-like receptor; Ac, acetyl.

## Data Availability

All data presented and discussed are contained within the manuscript.

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
