# Peer review of "NOX4 Mediates Pseudomonas aeruginosa-Induced Nuclear Reactive Oxygen Species Generation and Chromatin Remodeling in Lung Epithelium"

_antioxidants, 2021, doi:10.3390/antiox10030477_

Round 1

Reviewer 1 Report

The manuscript of Fu et al. investigates a novel aspect of lung epithelial cells NOX4-derived ROS production in the lung injury imposed by Pseudomonas Aeruginosa (PA) infection.  Authors attempt to demonstrate that activation of NOX4 ROS production is mediated through an isoform-specific upregulation sphingosine kinase 2 that would in turn induce NOX4 mRNA levels and promote H3 and H4 histone acetylation. To achieve their study, Authors use both an in vivomurine model (mice with lung epithelial cell-specific deletion of NOX4) and in vitro models (mouse lung and human primary bronchial epithelial cells). Authors conclude that in lung alveolar epithelial cells NOX4 enhances inflammatory gene expression upon PA infection through HDAC oxidation inducing its histone acetylase activity.

The manuscript is clearly written, easy to read and follows a logical train of thought. The manuscript is well in the scope of theme for the journal “Antioxidants”.

Major concerns:

  1. As a general remark as noted at several places below, Authors should follow the now widely accepted requirement of clearly marking Western blots with molecular weight markers and provide the original non-manipulated images for each gel presented in the Supplementary Figures. It is of special concern for blots of NOX4 as the reliability of diverse NOX4 antibodies concerning their specificity and isoform recognition is a constant source of debate in the NOX research field. Authors apply a NOX4 antibody that according to the supplier website documents shows a strong band in HEK293 cells. HEK293 cells are generally used as a negative control for the detection of NOX4 protein or activity (eg. Zhang, Biochimie 93 (2011) 457e468, doi:10.1016/j.biochi.2010.11.001 457e468; or Prior, JBC,2016, Vol 291, No 13, pp.7045-7059). This reviewer does not call into doubt the NOX4 blots presented in the manuscript but finds is imperative to provide unmodified blots for clarity.

Figure 1 G and H it seems that there were only 3 mice/group measured, in contrast to the other panels in the figure where 5 mice seem to be noted.

Figure 2 A images of colocalization between NOX4 and SPC are not convincing, please provide better quality and higher magnification images.

Figure 2 C No molecular weight of the detected bands are shown; please provide uncut whole blots with molecular weight markers in Supplementary materials.

Figure 1 D and Figure 2 C Please explain the observed approximately 3-4X increase in BAL H2O2 levels in PAchallenged NOX4 Spc-Cre mice compared to the vehicle-treated mice (Figure 1 D). Even though the levels of H2O2 are lower in PA challenged NOX4 Spc-Cre mice compared to the NOX4fl/fl mice they are still elevated compared to the levels observed in the non-challenged state.

Figure 3 A Please explain the lack of stimulatory effect of PA on NOX4 expression on alveolar Type II cells isolated from NOX4 flox/flox mice (compare to the image of the effect of PA on whole lung tissue in Figure 2 C). Could it be that the major cell type responding to PA infection in terms of stimulation of NOX4 expression is other than the alveolar cells?

Figure 3 A As in other figures, the exact molecular weight of the bands detected are crucial, especially that in isolated cells the NOX4 antibody seems to detect a double band compared to the lung tissue (Figure 2 C).

Figures 3 A and B follow different orders in the presentation of Western blots, (A) starts with NOX4 while (B) presents NOX4 at the bottom of the panel. It would be desirable to follow the same pattern for better flow of thought.

Figure 3 C Translocation of NOX4 into the nucleus is a crucial and important observation of the study. Again, it would be necessary to label the blots and provide the whole uncut blot in the Supplementary material for each panel presented in the Figure.

Figure 3 D Which NOX4 band did the Authors quantify (lower or higher or the two bands together?). It is a crucial question as little information is available concerning the roles of diverse NOX4 isoforms in pathophysiological settings. Authors should attempt to run gels with the same samples to be able to better distinguish the two bands recognized by the NOX4 antibody in the cytoplasm and the nucleus.

Figure 4 A and B Authors provide the reference for the methods applied for the lung instillation of NOX4 and NOX2-specific siRNAs. To provide proof of the specific effects of the siRNAs applied in the current study Authors should show immunohistochemical images of NOX2 derived from NOX4 siRNA-treated mice and vice versus. Similarly, such “cross-over” Western blots and their quantifications should be provided in panel B. 

Figure 6A clearly demonstrates the upregulation of nuclear ROS production in the “merged” image of PA-stimulated cells. It is not clear for this reviewer why such merged images are not provided in Figure 5 C where the images of the PA-stimulated scrambled and NOX2-specific siRNA transfected cells are clearly different colors from Figure 6A.

Figure 6A DAPI staining in the GKT-treated cells seems to be of much lower intensity compared to the non-treated cells. Were these images taken with the same microscope settings as the other images in the panel? If yes, can authors provide a possible explanation for this difference in staining intensity?

Figure 7 A, C and E It is not clear for this reviewer why DAPI staining is not employed in these images. DAPI staining would provide the necessary confirmation about the presence of cells in the images. Authors should provide these images. 

Figure 7 B, D and F The “y” axis of the graph is marked as Nuclear P-Hyper Fluorescence Intensity. How are the quantifications presented in these graphs are different from the ones presented in Figure 5B and D as well as Figure 6 B where the “y” axis is entitled Nuclear ROS Intensity? 

Figure 8 B, D and F Please refer to my previous comments about the presentation of Western blots and the necessity of providing unmanipulated versions for the Supplementary Material.

Author Response

We thank the reviewer for the insightful and positive comments regarding the manuscript. The comments and suggestions have been considered and addressed.

Major concerns:

Comment 1: As a general remark as noted at several places below, Authors should follow the now widely accepted requirement of clearly marking Western blots with molecular weight markers and provide the original non-manipulated images for each gel presented in the Supplementary Figures.

Response: As per requirement of the journal and reviewer’s comment, all the Western blots have been marked with molecular weight markers in the revised manuscript. Further, the original non-manipulated film images for each gel are provided as supplemental material.

Comment 2: It is of special concern for blots of NOX4 as the reliability of diverse NOX4 antibodies concerning their specificity and isoform recognition is a constant source of debate in the NOX research field. Authors apply a NOX4 antibody that according to the supplier website documents shows a strong band in HEK293 cells. HEK293 cells are generally used as a negative control for the detection of NOX4 protein or activity (eg. Zhang, Biochimie 93 (2011) 457e468, doi:10.1016/j.biochi.2010.11.001 457e468; or Prior, JBC,2016, Vol 291, No 13, pp.7045-7059). This reviewer does not call into doubt the NOX4 blots presented in the manuscript but finds is imperative to provide unmodified blots for clarity.

Response: The reviewer’s concern is fully appreciated. We have provided unmodified blots of NOX4 from lung tissues (Fig. 2) and alveolar epithelial cells isolated from Nox4flox/flox and tamoxifen-induced Nox4flox/flox SPC-Cre mice (Fig. 3A). Further, the validity of the NOX4 antibody is shown in Fig. 3A, and Fig. 3B wherein NOX4 was genetically deleted in alveolar epithelial cells in vivo using tamoxifen-inducible SPC-Cre mice, and in vitro using scrambled and Nox4 siRNA in lung epithelial cells (Fig. 3B). 

Comment 3: Figure 1 G and H it seems that there were only 3 mice/group measured, in contrast to the other panels in the figure where 5 mice seem to be noted.

Response: Thanks for pointing out this error in Fig. 1 G & H., which has been corrected to indicate five, and not three, mice in the revised manuscript.  

Comment 4: Figure 2 A images of colocalization between NOX4 and SPC are not convincing, please provide better quality and higher magnification images.

Response: As suggested, a higher magnification and better-quality image is provided for Fig. 2A with the type 2 pneumocytes labelled, in the revised manuscript. Further, we had inadvertently labeled top of the images in the previous version NOX (red) and SPC (green) though the figure legends had the correct labels assigned. This has been corrected in the revised Fig. 2 where NOX4 is labelled green and SPC red on the top of the images.

Comment 5: Figure 2 C No molecular weight of the detected bands are shown; please provide uncut whole blots with molecular weight markers in Supplementary materials.

Response: The molecular weights of the bands are marked and uncut whole blots are provided as supplementary material.

Comment 6: Figure 1 D and Figure 2 C Please explain the observed approximately 3-4X increase in BAL H2O2 levels in PA challenged NOX4 Spc-Cre mice compared to the vehicle-treated mice (Figure 1 D). Even though the levels of H2O2 are lower in PA challenged NOX4 Spc-Cre mice compared to the NOX4fl/fl mice they are still elevated compared to the levels observed in the non-challenged state.

 Response: This is a relevant point raised by the reviewer. Infection of mouse lung with Pseudomonas aeruginosa (PA) increases ROS production (including hydrogen peroxide) via NOX2 and NOX4 (Fu et la., AJRCMB 48: 477-488, 2013; cited Ref 15). In this study, NOX4 we have deleted NOX4 in the alveolar epithelial cells and NOX2 is still stimulated by PA resulting in considerable ROS production. Further, in addition to alveolar epithelial cells, mouse lung endothelial cells also express NOX2 and NOX4 (Pendyala S. et al., Antioxidant Redox Signaling 11: 747-764, 2009; cited Ref #24). Thus, NOX2 in the alveolar epithelial cells and NOX2/NOX4 in lung ECs could be contributing to ROS production that could explain the elevated ROS in alveolar epithelial cell NOX4 KO mice. To make this clear, we have added additional sentences in the discussion section of the revised manuscript.

Comment 7: Figure 3 A Please explain the lack of stimulatory effect of PA on NOX4 expression on alveolar Type II cells isolated from NOX4 flox/flox mice (compare to the image of the effect of PA on whole lung tissue in Figure 2 C). Could it be that the major cell type responding to PA infection in terms of stimulation of NOX4 expression is other than the alveolar cells?

Response: As pointed out by the reviewer, NOX4 expression in Fig. 2A and Fig. 3A has been determined under different conditions. In Fig. 2C, NOX4 expression was determined in total lung lysates after 24 h of PA infection of mouse lung while in Fig. 3A, isolated alveolar type II epithelial cells were exposed to vehicle or heat-inactivated PA for 3 h and not 24 h. Hence, there is a difference in the exposure time with PA, in vivo and in vitro, which most likely accounts for the difference in expression level of NOX4. However, we have not done a detailed analysis of NOX4 expression in different cell types in the mouse lung between 0-24 h post-PA infection.

Comment 8: Figure 3 A.  As in other figures, the exact molecular weight of the bands detected are crucial, especially that in isolated cells the NOX4 antibody seems to detect a double band compared to the lung tissue (Figure 2 C).

Response: This is a valid point and as suggested the molecular weight of the bands detected are provided not only in Fig. 3A but all the Western blots of the revised manuscript. Also Figure 3 has been rearranged for ergonomic reasons.

Comment 9: Figures 3 A and B follow different orders in the presentation of Western blots, (A) starts with NOX4 while (B) presents NOX4 at the bottom of the panel. It would be desirable to follow the same pattern for better flow of thought.

Response: This is a useful suggestion; however, in this figure the emphasis is on histone acetylation after knocking down Nox4 with siRNA. Hence, we opted to not to change the presentation order as suggested.   

Comment 10: Figure 3 C Translocation of NOX4 into the nucleus is a crucial and important observation of the study. Again, it would be necessary to label the blots and provide the whole uncut blot in the Supplementary material for each panel presented in the Figure.

Response: The blot was labeled with molecular weight marker and uncut blot is provided in the supplemental material.

Comment 11: Figure 3 D Which NOX4 band did the Authors quantify (lower or higher or the two bands together?). It is a crucial question as little information is available concerning the roles of diverse NOX4 isoforms in pathophysiological settings. Authors should attempt to run gels with the same samples to be able to better distinguish the two bands recognized by the NOX4 antibody in the cytoplasm and the nucleus.

Response: The two bands run very closely and based on the running conditions of the 10% gel, the separation is not seen in all the blots. Therefore, wherever applicable, both the bands were quantified together and shown in Fig. 3C & D.

Comment 12: Figure 4 A and B Authors provide the reference for the methods applied for the lung instillation of NOX4 and NOX2-specific siRNAs. To provide proof of the specific effects of the siRNAs applied in the current study Authors should show immunohistochemical images of NOX2 derived from NOX4 siRNA-treated mice and vice versus. Similarly, such “cross-over” Western blots and their quantifications should be provided in panel B.

Response: These experiments were done at least 2-3 years ago and due to technical reasons, the stored lung tissues were not satisfactory for any further analysis. Therefore, we have deleted the data of NOX2 and NOX4 siRNA in the revised manuscript (Fig 4 is now deleted). 

Comment 13: Figure 6A clearly demonstrates the upregulation of nuclear ROS production in the “merged” image of PA-stimulated cells. It is not clear for this reviewer why such merged images are not provided in Figure 5 C where the images of the PA-stimulated scrambled and NOX2-specific siRNA transfected cells are clearly different colors from Figure 6A.

Response: As suggested we are providing merged images of the PA-stimulated ROS generation in scrambled, and Nox2 and Nox4 siRNA treated cells in the revised manuscript (New Figure # 4).

Comment 14: Figure 6A DAPI staining in the GKT-treated cells seems to be of much lower intensity compared to the non-treated cellsWere these images taken with the same microscope settings as the other images in the panel? If yes, can authors provide a possible explanation for this difference in staining intensity?

Response: All the images were taken at the same time and with the same setting using high resolution microscopy that captures the staining in a very thin layer. However, the merged images shown in Fig. 5 (previous version Fig 5; New Fig. 4) and Fig. 8 (new version) were captured using a regular fluorescent microscope. This is a possible reason for differences in staining intensity.

Comment 15: Figure 7 A, C and E It is not clear for this reviewer why DAPI staining is not employed in these images. DAPI staining would provide the necessary confirmation about the presence of cells in the images. Authors should provide these images.

Response: Sorry, we did not perform DAPI staining for the nuclear ROS images shown in Fig. 7.   

Comment 16: Figure 7 B, D and F The “y” axis of the graph is marked as Nuclear P-Hyper Fluorescence Intensity. How are the quantifications presented in these graphs are different from the ones presented in Figure 5B and D as well as Figure 6 B where the “y” axis is entitled Nuclear ROS Intensity?

Response: Nuclear ROS generation was measured by transfecting the cells with a nuclear-P-HyPer plasmid. “Y” axis representation of Nuclear p-HyPer Fluorescence Intensity and Nuclear ROS Intensity are the same and were quantified by life imaging using confocal microscope. To keep the ‘Y” axis same, we have changed the representation to “Nuclear ROS Intensity”.

Comment 17: Figure 8 B, D and F Please refer to my previous comments about the presentation of Western blots and the necessity of providing unmanipulated versions for the Supplementary Material.

Response: We have included the unmanipulated scans of the blots in the supplemental material file.

Reviewer 2 Report

Authors showed the important role of NOX4 in lung epithelial cell during P. aeruginosa (PA) infection. These results are very interesting. However, I have several questions about this study.

Major comments:

Innate immunity is the first and most important biological defense system against infectious diseases. It is difficult to believe that deletion of only one protein in the lung epithelial cell strongly attenuate PA-induced damages. At PA infection to the lung, neutrophiles increase in the lung and kills PA (https://doi.org/10.4049/jimmunol.175.6.3927). Why did NOX4 deletion attenuate innate immunity? In this study, tamoxifen is used to generate NOX4 deletion. Tamoxifen itself can reduce lung neutrophile (DOI: https://doi.org/10.1101/2020.06.19.161919,10.1016/j.rvsc.2016.11.003). Authors need to exam the tamoxifen effect against PA infection by using wild type mice.

There is no data about NOX4 siRNA and inflammation. But those were in your previous paper (reference 18). In the paper, the efficacy of NOX4 siRNA was about a half and NOX4 siRNA did not reduce inflammation. Those data are inconsistent with this paper's NOX4 deleted mice data. These differences may be caused by tamoxifen. In vivo experiments using GKT137831 ± PA may also be able to show the direct effect of NOX4 against PA infection.

IL-6 is mainly generated from macrophage during infection. In vitro experiment ± heat killed PA, did you measure IL-6 and/or TNF-α in the medium? Did you find direct relation between histone acetylation and IL-6 and/or TNF-α?

Minor commnents:

There is no detail information about PA.

“P. aeruginosa” should be italic.

If your schema in Fig 10 is correct, why RC1 inhibitor reduced H2O2 level?  H2O2 increase is caused by NOX4 activation, and RC1 is relate to the next step.

Figure 1 study, did PA present in the lung? Fig. 1A image is good but magnified images help readers’ understanding.

NOX4 IHC data are unclear (Fig.2A, 4A). Fig. 2A is too weak, Fig. 4A is too strong. Fig. 4A should be presented with more magnified images and staining NOX2 or NOX4 should be specified.  

PA flagella are present only on one side. Fig. 10 PA image is not good.

Figure 2C, Ac-H3, H3 image should be deleted. Same images are in fig3A.

Author Response

We thank the reviewer for the positive comments regarding this study.

Major comments:

Comment 1: Innate immunity is the first and most important biological defense system against infectious diseases. It is difficult to believe that deletion of only one protein in the lung epithelial cell strongly attenuate PA-induced damages. At PA infection to the lung, neutrophiles increase in the lung and kills PA (https://doi.org/10.4049/jimmunol.175.6.3927). Why did NOX4 deletion attenuate innate immunity?

Response: Thanks for raising this important and interesting point regarding PA infection of the lung, neutrophil infiltration, and innate immunity. PA infection of the lung increases neutrophil infiltration and through various mechanisms, kills the invading bacteria. However, uncontrolled influx of neutrophils can also damage the lung tissue and delay the repair of lung injury induced by PA. Thus, there is a balance between neutrophil infiltration, bacterial killing and resolution of injury and repair. The published work by Ramphal R et al, (J Immunology 175: 3927-3934, 2005) suggest that the recognition of LPS by TLR2 or TLR4 is not central for the susceptibility to PA mediated lung infection. Our studies related to epithelial NOX4 and PA induced lung inflammatory injury involve flagellin signaling via TLR5 and generation of nuclear ROS in lung epithelial cells (see schema in New Fig 9). We have observed that flagellin, but not LPS, mediates nuclear ROS generation (data unpublished) and hence there are at least two distinct pathways (TLR2/TLR4) and (TLR5) that are involved in PA-induced lung inflammation, injury and resolution. In our hands, knockdown of NOX4 in alveolar lung epithelial cells partially reduced IL-6 and TNF-alpha levels in the BAL fluid suggesting persistence of innate immune response at 24 h post PA challenge. We and others have shown that mouse lung clears PA 6-10 h post-infection although the innate immune responses are seen for several days.

Comment 2: In this study, tamoxifen is used to generate NOX4 deletion. Tamoxifen itself can reduce lung neutrophile (DOI: https://doi.org/10.1101/2020.06.19.161919,10.1016/j.rvsc.2016.11.003). Authors need to exam the tamoxifen effect against PA infection by using wild type mice.

Response: Thanks for bringing up the issue of the side effect of tamoxifen itself. To address this, NOX4 flox/flox mice, which served as the controls, were administered with the same dose and for identical duration of tamoxifen along with the NOX4 flox/flox – SPC-Cre mice. We did not include regular wild type mice as tamoxifen controls due to large animal numbers from the various groups with and without PA challenge. This has been clearly stated in the Experimental Section. 

Comment 3: There is no data about NOX4 siRNA and inflammation. But those were in your previous paper (reference 18). In the paper, the efficacy of NOX4 siRNA was about a half and NOX4 siRNA did not reduce inflammation. Those data are inconsistent with this paper's NOX4 deleted mice data.

Response: This is an important point. As the Nox4 siRNA experiment does not add any additional value to the NOX4 KO data, we have deleted the Nox4/Nox2 siRNA section in the revised manuscript (Fig 4 deleted in the revised version).

Comment 4: These differences may be caused by tamoxifen. In vivo experiments using GKT137831 ± PA may also be able to show the direct effect of NOX4 against PA infection.

Response: This is an excellent suggestion. However, this and other experiments related to RAC1 are beyond the scope of the current study but will be considered for future investigations.

Comment 5: IL-6 is mainly generated from macrophage during infection. In vitro experiment ± heat killed PA, did you measure IL-6 and/or TNF-α in the medium? Did you find direct relation between histone acetylation and IL-6 and/or TNF-α?

Response: This is another important point. We have partly addressed this in our earlier publication (Ebenezer DL et al., Thorax 74: 579-591, 2019) wherein IL-6 levels were measured in lung epithelial cells treated with heat-inactivated PA under  conditions of blocking SPHK2, PKC δ and histone acetylation. However, a similar study in the context of NOX4 inhibition has not been performed.

Minor comments:

Comment 1: There is no detail information about PA.

Response: This information is added.

Comment 2: “P. aeruginosa” should be italic.

Response: This has been carried out.

Comment 3: If your schema in Fig 10 is correct, why RC1 inhibitor reduced H2O2 level?  H2O2 increase is caused by NOX4 activation, and RC1 is relate to the next step.

Response: Thanks for pointing this out. Our data suggest that blocking Rac1 reduced PA-induced nuclear ROS production and PKC δ-siRNA treatment attenuated Rac1 activation by PA. We believe that Rac1 activation is downstream of PKC δ and SPHK2/S1P signaling in the nucleus, but upstream of NOX4. Accordingly, changes have been made to the schema (Fig. 9) depicting activated Rac1 upstream of NOX4.

Comment 4: Figure 1 study, did PA present in the lung?

Response: The mice were infected with PA and terminated at the end of 24 h. We have shown earlier that at the end of 6 h of post-infection, the CFU in lungs are very low (Ebenezer DL et al., Thorax 74: 579-591, 2019).

Comment 5: Fig. 1A image is good but magnified images help readers’ understanding.

Response: Not sure about this comment

Comment 6: NOX4 IHC data are unclear (Fig.2A, 4A). Fig. 2A is too weak.

Response: Fig 2 A has been changed to a larger magnification in the revised manuscript with the Type II pneumocytes labelled.

Comment 7:  Fig. 4A is too strong. Fig. 4A should be presented with more magnified images and staining NOX2 or NOX4 should be specified.  

Response: This Figure and data have been eliminated in the revised version.

Comments 8: PA flagella are present only on one side. Fig. 10 PA image is not good.

Response: This has been modified.

Comment 9: Figure 2C, Ac-H3, H3 image should be deleted. Same images are in fig3A.

Response: Thanks for pointing out, there was an error in image compilation. Ac-H3 and H3 images have been deleted and replaced by actin loading controls in the revised Figure 2C. The originals blots are provided in the supplementary section.

Round 2

Reviewer 2 Report

Same as previous comment 5, there is no direct relation of NOX4 and IL-6 and/or TNF-α expression. I  read the previous paper (Ebenezer DL et al., Thorax 74: 579-591, 2019). It was similar, but key molecule is different from this study. If you present the schema (figure 9), IL-6 and/or TNF-α should be determined. 

Author Response

Comment: Same as previous comment 5, there is no direct relation of NOX4 and IL-6 and/or TNF-α expression. I read the previous paper (Ebenezer DL et al., Thorax 74: 579-591, 2019). It was similar, but key molecule is different from this study. If you present the schema (figure 9), IL-6 and/or TNF-α should be determined. 

Response: We thank the reviewer for this critique. We concur with the reviewer that the in vitro study did not directly demonstrate the relationship between NOX4 and IL-6/TNF alpha as depicted in the schema Fig. 9 and remains to be established. Accordingly, we have modified the schema (new Fig 9) in the revised version of the manuscript, wherein the effect on transcriptional regulation of inflammatory genes is depicted as dotted line. The figure legend has also been revised to indicate this.